# Object Selection as a Biometric

**DOI:** 10.3390/e24020148

**Published:** 2022-01-19

**Authors:** Joyce Tlhoolebe, Bin Dai

**Affiliations:** 1School of Information Science and Technology, Southwest Jiaotong University, Chengdu 610031, China; daibin@home.swjtu.edu.cn; 2The State Key Laboratory of Integrated Services Networks, Xidian University, Xi’an 710071, China; 3Peng Cheng Laboratory, Shenzhen 518055, China

**Keywords:** biometric authentication, biometrics, eye tracking, pattern recognition, saccades, machine learning

## Abstract

The use of eye movement as a biometric is a new biometric technology that is now in competition with many other technologies such as the fingerprint, face recognition, ear recognition and many others. Problems encountered with these authentication methods such as passwords and tokens have led to the emergence of biometric authentication techniques. Biometric authentication involves the use of physical or behavioral characteristics to identify people. In biometric authentication, feature extraction is a very vital stage, although some of the extracted features that are not very useful may lead to the degradation of the biometric system performance. Object selection using eye movement as a technique for biometric authentication was proposed for this study. To achieve this, an experiment for collecting eye movement data for biometric purposes was conducted. Eye movement data were measured from twenty participants during choosing and finding of still objects. The eye-tracking equipment used was able to measure eye-movement data. The model proposed in this paper aimed to create a template from these observations that tried to assign a unique binary signature for each enrolled user. Error correction is used in authenticating a user who submits an eye movement sample for enrollment. The XORed Biometric template is further secured by multiplication with an identity matrix of size (*n* × *n*). These results show positive feedback on this model as individuals can be uniquely identified by their eye movement features. The use of hamming distance as additional verification helper increased model performance significantly. The proposed scheme has a 37% FRR and a 27% FAR based on the 400 trials, which are very promising results for future improvements.

## 1. Introduction

Security and privacy issues are common problems in the field of computer science, and they are a general national concern. Privacy is contextually based on the specifics of who should do what and where [1]. Recently, the availability of low-cost data that aids in ample data communication, securing data and applications from unauthorized identities is becoming very important to help control access to software, data, devices and physical buildings [2]. Despite the widespread use of passwords to authenticate users, there are many weaknesses and drawbacks associated with this method of user authentication [3]. For example, in a password authentication system, there is a verification table used to verify the users’ passwords when they log in to the system [3]. If an event of a security breach occurs and the verification table is compromised by an intruder, then the whole system or parts of the system will be breached.

However, alphanumeric passwords are vulnerable to shoulder surfing attacks, dictionary attacks and social engineering attacks [4]. Shoulder surfing is whereby someone who is not intended to know the user’s password can look over the user’s shoulder as they enter the password and thereby obtain the password [5]. Social engineering attacks are based on human interaction, and they normally involve tricking users to break security policies because they appear harmless and look appropriate [6]. For example, in a social networking site such as Facebook, the attacker can analyze and exploit the user’s interest and send some recommendations based on the information that was gathered from their profile [7].

## 2. Biometric Authentication

As one of the countermeasures to the problems encountered with the use of passwords, the adoption of biometric authentication systems emerged. Biometric authentication is a system whereby a user is identified or verified with their physical or behavioral characteristics such as their iris pattern, DNA, fingerprints, voice and face [8]. The use of biometrics is growing rapidly and is used for authentication in applications such as border controls and access control to physical facilities [8,9]. This method of user authentication can also be used even in setting up profiles on mobile devices [8]. The use of biometric authentication methods developed as one of the solutions used to counter problems with traditional authentication methods such as the password. A significant growth in the market for biometric authentication is expected, from $2.0 billion in 2015 to $14.9 billion by 2024 [10]. However, in real applications, there are still some security concerns encountered with current biometric authentication methods [11]. It is possible for intruders to breach the security of the database storing the biometric information and steal it [10].

Eye movement as a biometric technique makes use of physiological and behavioral aspects, and these traits make this biometric difficult to copy [11,12,13] as compared to traditional authentication techniques. Eye movement biometrics identifies behavioral patterns and other information regarding physiological properties that generates eye movements [14]. Several studies showed that the use of eye movement for human identification is a promising research field [12,13,14]. In previous studies, a high level of accuracy has been achieved, such that eye movement biometric can be an independent biometric technique without being integrated with other modalities [14]. However, recognition error rates for this technique are still high, but recent progress has been promising [15]. The usability and acceptability of eye movement as a biometric technique can be highly influenced by the overall design of the interfaces and the objects used in extracting the eye movement pattern [16]. The movement of eyes can be obtained from observing images, reading of text, jumping points pattern, or any vision process, and this movement can be recorded using an eye-tracking device [13,17].

Research on eye movement as a biometric was first done in 2004 by Kasprowsi et al., where they designed a jumping point as their object of observation [15,18,19,20]. The jumping point was displayed in a computer screen to jump to different points of the screen, and they analyzed eye movement pattern of users as they watched the jumping point on the screen. The overall recordings of the movement patterns results were obtained from measuring features such as “…distance to stimulation, eye difference, Fourier and Wavelet transform of the eye signal…” [19]. A 16% average error rate was obtained from this method after the tests were done several times for each user [19]. The results of this method opened a research path for many researchers to carry out some investigations on this method but using different objects to carry out their experiments [18]. Instead of using one object as the stimulus, a study by [17] went further by doing experiments on different objects and combining them together. The experiment included presenting users with images of human faces followed by a white screen with a small cross at the center, a moving circle and black and white shapes that were randomly presented to record the movement of their eyes. Hotspots were also able to be recorded as they showed the highest concentrations of fixations as the user observed the objects mentioned above.

## 3. Acceptance and Privacy Issues of Biometric Systems

Although some biometric systems may provide high levels of security, the targeted group may not accept it, depending on the issues surrounding the privacy of users. A biometric system that is rejected by a large number of the targeted population is not suitable for use [21]. If a biometric system is comfortable and ease to use, that will highly contribute to its acceptance [22]. Users prefer little interaction with the biometric system, and they will deem it as being useable and more convenient for them. However, as a measure to minimize user interactions with the biometric systems, there are some biometric systems that are able to identify users and extract other additional information without the users’ knowledge, and this has raised some privacy issues [22,23]. For example, biometric-based recognition systems such as the retina pattern recognition system may provide some additional background information about the health of the users’ eyes, which therefore may deprive the users of this information [22]. As a second example, in a fingerprint-recognition system, if a person has some malformed fingers, this might be statistically associated with some certain genetic disorders [23].

However, the main issues on biometric systems is that biometric systems are associated with the conflict on how the data will be collected, protected and used, as well as the conflict with each individual’s beliefs and values [24,25]. The flexibility of data movement on the internet also leads to loss of data control from user [13]. Some biometric identification systems such as the fingerprint systems may lead to the possibility of unwanted identifications of individuals since some systems have recently become centralized [21,23]. Weaknesses in the use of traditional methods have led to the evolution on the use of biometric authentication methods that use the “what you have” technique. This technique brought some advancement in security. These techniques include fingerprint, retina, iris, eye movement and face recognition. The use of a particular biometric technique depends on what you want the system to achieve. Studies have shown that using one or more biometric techniques can improve security. Eyes have features that produce useful information that can be used in biometric authentication. While different studies in to the general applicability of eye movement biometrics has been done, some of the proposed approaches had some high error rates and low identification rates. In this work, we consider measuring various characteristics of the eye movement pattern (the scan path) and analyzing those characteristics to uniquely identify individuals.

## 4. Materials and Methods

### 4.1. Participants

Data were collected from a total number of 20 participants. Participants were randomly personally invited and some were invited by email, and they were from different departments in the university. The age of the participants ranged from 18 to 38. The minimum age of participants was 18 years. From the targeted participants, all participants managed to engage to the end.

### 4.2. Tasks

The experiment consisted of one task, which involved choosing and finding objects displayed in 30 slides. All the slides displayed 15 different objects of the same size and different colors and all were equidistant with the same properties (feel and design). On the first slide, participants were asked to scan the objects displayed on the screen. The first slide was displayed for 30 s to give participants enough time to learn the objects, and the rest of the slides were each displayed for eight seconds. On the second slide until the last one, objects were rea-arranged, and participants were tasked to choose any three objects of their choice. Re-arranging objects on the other slides was done to encourage participants to be fully engaged in the experiment as they were required to always look for the three objects that they chose.

### 4.3. Equipment and Software

The device used to record eye movements was the Senso Motoric Instruments (SMI) RED250 mobile eye tracker operating with a temporal resolution of 250 Hz. The objects used in the experiment were all designed with Microsoft PowerPoint 2013. The objects (stimuli) were presented on a white background flat screen monitor in a screen resolution of 1920 × 1080 pixels. The eye tracker was placed on a table and the objects (stimulus) were at a rough distance of approximately 50 cm from the participant’s eyes. The software was able to give feedback during the calibration phase. This distance between the participant’s eyes and the eye tracker was fixed with a 10 cm flexibility range due to differences in participants’ body structure. Some were tall, and some were short and therefore the distance could not be strictly fixed to the same distance. The eye tracking equipment was installed with an eye positioning software that was able to guide the participants to position their eyes within the scene area. A red arrow blinked as an indication to show the participants the direction in which they should try to position their eyes. Data were analyzed using the Be Gaze 3.6.52 analysis software which is part of the former SMI (Senso Motoric Instruments) originally from Paris, France.

### 4.4. Feature Extraction and Selection

A complete scan path is described by its particular characteristics, and those characteristics can be used to uniquely identify individuals [12]. Figure 1 below shows a random selection of users scan paths. From the picture we could tell an object with the highest fixation duration because it has a bigger circle around it. Eye movement tracking equipment is a sophisticated tool that is able to measure a variety of properties of eye movement patterns. For this study, only certain properties that were needed were recorded and used. Even if the pattern can be the same, some properties of the patterns are completely different because physiological actions are influenced by brain activity [14]. The following metrics were measured and used for unique personal identification [12]: fixation duration, fixation count, scan path length, fixation frequency, saccade frequency count, saccade duration, saccade velocity and the object fixation duration. These metrics were measured based on each scan path for each individual.

Fixation duration: this is the time in seconds that shows how long a participant took to fixate in the experiment.

Fixation Count: this is the total number of fixations the participants made in the entire experiment recording [12].

Dwell time: this is the time the participant spent looking within the boundary of an object.

Average blink Duration: this is the average number of blinks a participant made throughout the experiment.

Object fixation Duration: this is time in which participants fixated on particular objects. Duration of fixation on each object on the scene will be calculated. See appendix 3 for each object fixation duration.

Scan path length: this feature can be calculated by measuring the total distance between all the fixation points in a scan path [12].

Fixation frequency: this is the time in seconds in which the participant fixated on a particular object.

Saccade Amplitude average: A saccade is the movement of eyes between the fixations. The saccade amplitude is obtained by summing the horizontal and vertical saccade amplitudes and dividing them by the number of saccades.

Pupil diameter: this is the size of the pupil and is normally computed by the eye-tracking equipment. The average pupil size varies from 2 to 4 mm in diameter and it normally increases from about 4 mm in bright light and 8 mm in the dark [26].

Saccade frequency count: this is the speed in which the eyes move from one fixation point to the other.

Saccade duration average: this is the average time in milliseconds taken by an individual to complete the saccade [26]. Saccades are in most cases completed within an average time of ten milliseconds.

Saccade Velocity: the saccade velocity is the speed in which a saccade was completed.

## 5. Architecture of the Classification Process

From the 600 observations, 300 were used for enrollment and the other 300 were used as testing data. Figure 2 below shows the classification process which shows how data was analyzed. Cross validation was done by randomly selecting a sample from the entire sample except the instances defined in creating the template for enrollment in that specific trial., e.g., if the first 30 observations (3 realizations for a single user) are used to create user 1 biometric template then the cross-validation set was defined as a random trial from all other observations except the 30 that are used. The testing dataset contained only 1 realization for each user with the assumption that only 1 sample will be accepted and recognized as the attempt biometric.

### 5.1. Classification Method

Let *x*_1_, …, *x_n_* be a correctly classified sample in classes *θ*_1_, …, *θ*_2_, where *x_i_* takes values in a metric space upon which a distance function d is defined. We consider the pairs (*x_i_*, *θ^i^*) where the *x_i_*’s represent the p-variate observation on the *i*’th individual and *θ^i^* is the class to which they belong. It is usually said that *x_i_* belongs to *θ^i^*, which means that the *i*’th individual upon which measurements *x_i_* have been observed belongs to category $\theta^{i}\in \left\{\theta_{1},\ldots ,\theta_{M}\right\}$.

Consider a new pair of values (x,θ) where only ***x*** is observable and one wants to estimate the value of θ, using the training data that already have correctly classified observations. An observation
$$x^{\prime }\in \left\{x_{i,\ldots },x_{n}\right\}$$
is denoted the nearest neighbor of *x* if [27]
$$\underset{i=1,\ldots ,n}{\min}dist\left(x_{i},x\right)=dist\left(x^{\prime },x\right)$$

The NN classification method assign the observation *x* the category $\theta^{i}$ of its nearest neighbor $x_{i}$. The decision rule *k* has to be modified in the case of a tie between classes.

### 5.2. Distance Function

A case is classified by a majority vote of its neighbors, with the case being assigned to the class most common amongst its *k* Nearest neighbors measured by a distance function. If *k* = 1, then the case is simply assigned to the class of its nearest neighbor. The value of *k* is usually small and an integer with positive value. If *k* = 1, class allocation of the sample is based on the nearest one neighbor within a certain distance.

### 5.3. Euclidian Distance

This is an instance-based classification algorithm similar to the k-Nearest Neighbor (kNN). Each new instance ***x*** is compared with existing ones $x^{\prime }\in \left\{x_{i,\ldots },x_{n}\right\}$ using a distance metric, and the closest existing metric is used to assign a class $\theta^{i}$ to the new instance. The difference with this algorithm and the kNN is that instead of Euclidian distances, it uses the concept of entropy to define its distance measure. The Euclidean distance is a measure to find distance between two points. In Cartesian coordinates, if there are two points in Euclidian k-space, then the distance (*dist*) from/to is defined by Pythagoras’s theorem. The formula is as follows [28]:$$dist\left(x,y\right)=\sqrt{\overset{k}{\underset{i=1}{\sum }}{\left(x_{i},y_{i}\right)}^{2}}$$
where *a_i_* and *b_i_* are the values of the *i*’th argument of *x*, *y* vectors.

### 5.4. Naïve Bayes Classifier

This is a simple probabilistic classifier that applies the Bayes’ theorem with strong assumptions of independence. This method assumes that the presence of a particular feature of a class is unrelated to any other feature [29]. Parameter estimation is done using the method of maximum likelihood, and the Bayes classifier needs only a small sample of training data to estimate parameters.

### 5.5. Naïve Bayes Model

The probability model can be presented as a conditional probability model
$$P\left(C|F_{1,\ldots },F_{n}\right)$$
where *C* represents a dependent class variable with a small number of classes, and *F*_1_,…, *F_n_* are some feature variables. The model can be extended and presented such that [30]
$$P\left(C|F_{1},\ldots ,F_{n}\right)=\frac{P\left(C\right)P\left(F_{1,\ldots },F_{n}|C\right)}{P\left(F_{1,\ldots },F_{n}\right)}$$
which indicates that
posterior = (prior × likelihood)/evidence

The naïve assumption of conditional independence of features means that for features $F_{i}$ and $F_{j}$, there is conditional independence such that i≠j. Under the assumption of independence, the conditional distribution over the class variable *C* can be expressed as [30]
$$P\left(C|F_{1},\ldots ,F_{n}\right)=\frac{1}{Z}P\left(C\right)\overset{n}{\underset{i=1}{\prod }}P\left(F_{i}|C\right)$$
where *Z* is a scaling factor dependent only on $F_{i},\ldots ,\;F_{n}$. There are *k* classes, and in practice, classes are often binary (*k* = 2).

### 5.6. K-Star Algorithm

This is an instance-based classification algorithm similar to the k-Nearest Neighbor (kNN). Each new instance *x* is compared with existing ones $x^{\prime }\in \left\{x_{i,\ldots },x_{n}\right\}$ using a distance metric, and the closest existing metric is used to assign a class $\theta^{i}$ to the new instance. The difference with this algorithm and kNN is that instead of Euclidian distances, it uses the concept of entropy to define its distance measure.

### 5.7. Classification of K-Star

The classification with K-Star is made by summing the probabilities from the new instance to all of the members of a category. This must be done with the rest of the categories to finally select that with the highest probability [31,32]. The K-star algorithm uses entropic measures that are based on the probability of transforming an instance into another by choosing between all possible transformations using the random method. Using entropy as a measure for an instance distance is very beneficial. Information theory helps in computing the distance between the instances.

Let *I* be a set of instances that is possibly infinite and *T* be a finite set of transformations on *I*. Each T maps instances to instances as t:I→I. The transformations *T* contain a distinguished member denoted σ, which maps instances to themselves for completeness (σ(a)=a). Let *P* be the set of all prefix codes from *T**, which are terminated by σ (stop symbol). Members of *T** and P uniquely define a transformation on *I*.

The *K** function is defined as [33]
$$K^{*}\left(b|a\right)=-\log_{2}P^{*}\left(b|a\right)$$
where $P^{*}$, the probability function, is defined as the probability of all paths from distance *a* to distance *b*.

### 5.8. k Nearest Neighbor

*k* Nearest Neighbors is one of the simplest algorithms but it is very often accurate. Its main idea is that samples with the same classification tend to appear nearby. ‘Nearby’ refers to the distance between them being generally closer than the distance between samples with different classification. The distance may be calculated differently, but the Euclidean distance is used most frequently. Other distance measures include the Hamming distance and Manhattan/city block distance [30].
$$P\left(C|F_{1},\ldots ,F_{n}\right)=\frac{1}{Z}P\left(C\right)\overset{n}{\underset{i=1}{\prod }}P\left(F_{i}|C\right)$$
where *Z* is a scaling factor dependent only on $F_{i},\ldots ,\;F_{n}$. There are *k* classes and in practice classes are often binary (*k* = 2).

### 5.9. Naïve Bayes Classification

The Naïve Bayes classifier combines the probability model with a decision rule. The common approach is to choose the most probable hypothesis (called the maximum a posteriori or MAP decision rule). The classify function can be expressed as [34]
$$classify\left(f_{1,\ldots },f_{n}\right)=\arg\max P\left(C=c\right)\overset{n}{\underset{i=1}{\prod }}P\left(F_{i}=f_{i}|C=c\right)\;$$

### 5.10. Support Vector Machine

The support vector machine (SVM) is a supervised machine learning algorithm that can learn how to attach labels to objects [35,36]. The SVM uses a sample of objects (training) separated into classes to find the hyper plane in the data that produces the largest minimum distance between objects.

A linear SVM problem can be presented in terms of a two-class identification problem. The hyper plane decision boundary can be presented such that [35]
f(x)=β((w∗x)+b) 
where ‘*x*’ is the training data, ‘*b*’ is the bias vector and ‘*w*’ is the weight vector, and the norm of w determines the Vapnik–Chervonenkis (VC) dimensions.

Mapping is used by SVM schemes that are designed to ensure that dot products of pairs of input data vectors can easily be computed in terms of variables in the original space (training data set). This is done in terms of a kernel function *k(x*, *y*) selected to suit the problem.

### 5.11. J48

This is a widely algorithm used to construct decision trees. It is an implementation of C4.5 algorithm in Weka. Output for this algorithm is in the form of a decision tree with the roots that represent tests on an attribute and leaf nodes that represent classification [37]. An advantage of using this algorithm is that the results are easy to interpret because it works with nominal data and numeric data. A 98.51% accuracy was obtained by [37] when they compared the J48 with other decision tress under a 10-fold cross validation condition.

### 5.12. Statistical Presentation

Let the ‘*n*’ training vectors be denoted by $\left\{\left(x_{1},y_{1}\right),\left(x_{2},y_{2}\right),\ldots ,\left(x_{n},y_{n}\right)\right\}$ such that $x_{i}\in R^{n}$ and $y_{i}\in \left\{-1,1\right\}$. For linearly separable training data,
$$y_{i}\left(x_{i}*w+b\right)-1\geq 0\;;\;\mathrm{where}\;\forall i=1,\ldots ,n$$

The margin of points between two points in different classes is defined by two hyper-planes $\left(x^{*}w+b\right)=\pm 1$ such that no point can lie within this margin. Simplified calculation is done by converting the problem into the Lagrange framework and maximizing the value of the margin by minimizing the weights ‖w‖.

## 6. Results

From Table 1 below, the cryptosystem performs best where key size *k* = 7 returning the lowest FRR and FAR at 15% and 10%, respectively. A longer message length reduces the accuracy of the cryptosystem. The biometrics are binarized using the inter-class means difference from the intra-class means. This will return a zero when the column mean is less than the overall mean and a 1 otherwise. This helps to create an irreversible binary template and the initial binary data can be deleted. For experiment purposes, we have created a secret key of size 3 using a random number generator containing only ones and zeros [0,1]. This key/message is then run through BCH encoding after conversion to Galois field GF (2). The parameters for the BCH codes are (*k* = 3 and *n* = 9). The hash of the secret key is taken and stored later for user authentication. For the individual enrolling in the fuzzy commitment scheme, a secret key is further XORed with the biometric sample and the XORedbiometric template is further secured by multiplication with an identity matrix of size (*n* × *n*). The simulation was run 400 times on the first 10 subjects with 200 legitimate trials and 200 false inquiries. The model performed poorly in identification task as the BCH encoder tended to correct errors too well. The use of hamming distance as additional verification helper increased model performance significantly. The proposed scheme has a 37% FRR and 27% FAR based on the 400 trials during the identification process. The results suggest that the scheme would perform better if integrated with another biometric to improve its performance.

Figure 3 below shows a plot for the false rejection rate and the false acceptance rate for the biometric cryptosystem against the key size.

## 7. Analysis

The dwell-in time was used to identify the objects that were chosen by each participant. Any three objects with the highest dwell-in time were shown to be the participant’s choice from a list of the fifteen objects that were displayed. From a range of objects displayed on the scene, some objects attracted the participants’ attention more so than others. Although the influence of color in the selection of objects is not within the scope of this study, bright colored objects had an influence on the selection of objects. The book was chosen by 50% of the participants as one of their choice for the three objects, followed by the rose and the ball, consecutively.

The performance of the machine learning classifier has been assessed based on the accuracy, precision, true positive rate, false positive rate. and the Receiving Operating Characteristic curve (ROC). As shown in the results for the classifiers under different conditions, they performed slightly differently under different testing conditions. The Naïve Bayes classifier has always performed better regardless of the testing condition used. A 93% accuracy rate was obtained using the 10-fold cross validation with a 0.93 true positive rate. Table 2 below show the performance of different machine learning classifiers based on the tenfold cross validation.

## 8. Discussion

The experiment aimed to create a template from these observations that tried to assign a unique binary signature for each enrolled user. Error correction is used in authenticating a user who submits an eye movement sample for enrollment. Being able to uniquely identify individuals from a pool of users is an important aspect of security of computer systems and security in general.

Figure 4 below shows the enrollment, verification and validation process of users in to the biometric system.

Note that the permutation matrix proposed by [38] is not included in computation as a random key generation adds a heavy computational load to the system in trying to find an invertible (*n* × *n*) permutation matrix with an inverse.

For authentication/verification, a user *I* submits their eye movement sample, which is binarized using the mean as in the enrollment stage. This sample is drawn randomly from the joint feature vector matrix. For user 1, it is observations 1–30; user 2, observations 31–60; and so on. The helper data are made of the hash value of the encoded message and the variable ADS (User biometric + encoded message * identity matrix). ADS is multiplied with the inverse of identity matrix, and the query biometric sample is then added to obtain a corrupt version of the codeword through reverse arithmetic. The BCH decoder module is then used to decode the corrupt codeword and the hash function taken.

A comparison of the hash value with the one from enrollment opens for the legitimate user.

For additional protection, the hamming distance between the enrollment biometric template of and the query biometric template is used to aid decision by setting a threshold of 1. A legitimate attempt must have both of the following conditions: encoded codeword hash equal to decoded hash, and that maximum hamming distance must be 1 between query and enrollment binary templates.

Some previous works such as [15] measured the same eye movement metrics that we used in our study, but they did not use the same performance evaluation criteria, and therefore it is very difficult to compare the results of this study to other previous works. In comparison to the results to the results obtained in [20], an 83% accuracy rate was obtained during the classification process using the random forest method. Contrary to a study by [39], who conducted a study by measuring the saccadic eye movement signal from 109 young subjects, measured the users eye movement and focused only on measuring one scanpath feature, in our study, an 80–90% correct identification rate was obtained for saccadic eye movements. Based on the size of our data set, the chosen machine learning classifiers chosen were suitable for classifying the data. The number of tasks performed for this study could also be increased for comparison of the results. Collecting data from more participants could have been beneficial in terms of increasing the reliability of the presented results.

## 9. Conclusions

The main aim and objective of this study was to explore the potential of object selection using eye movement as an effective biometric. The results obtained from this study are very promising and show that it is possible to uniquely identify individuals by the unique features of their eye movement. The results show positive feedback on this study. The results from the machine learning classifiers showed a 93% accuracy rate. The proposed scheme has a 37% FRR and 27% FAR based on the 400 trials, which are very promising results. The results suggest that the scheme would perform better if integrated with another biometric to increase its performance. It is hoped that the ideas and results obtained from this study will advance and contribute to the understanding of eye movement as biometrics and provide some advancements in the eye-movement research field. Currently, the use or acceptance of eye movement biometric is relatively low as some advances in the emerging researches are still growing.

## Figures and Tables

**Figure 1 entropy-24-00148-f001:**
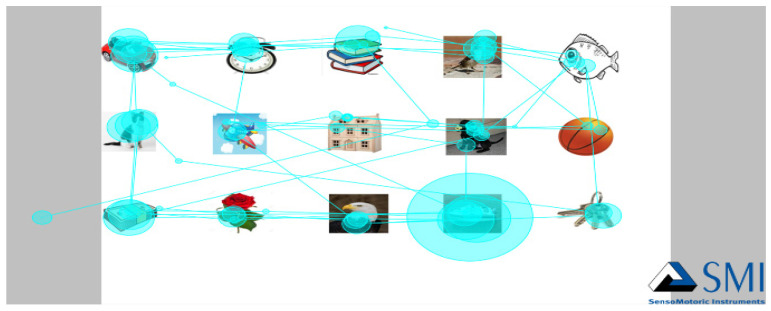
Scene Scan Path.

**Figure 2 entropy-24-00148-f002:**
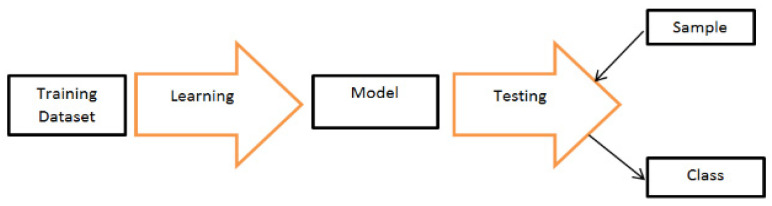
Architecture of the classification Process.

**Figure 3 entropy-24-00148-f003:**
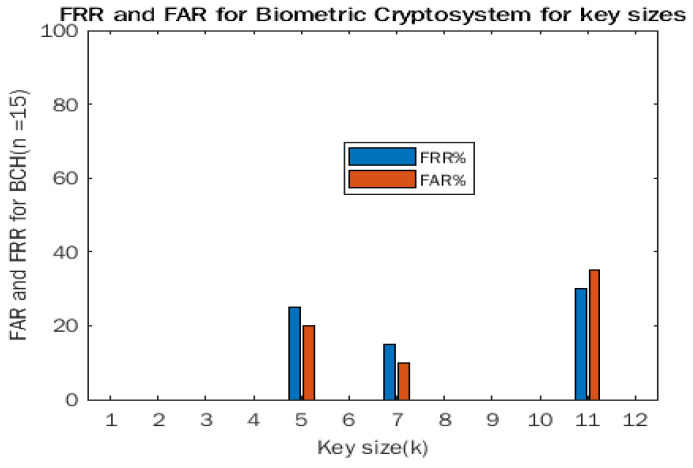
FRR and FAR.

**Figure 4 entropy-24-00148-f004:**
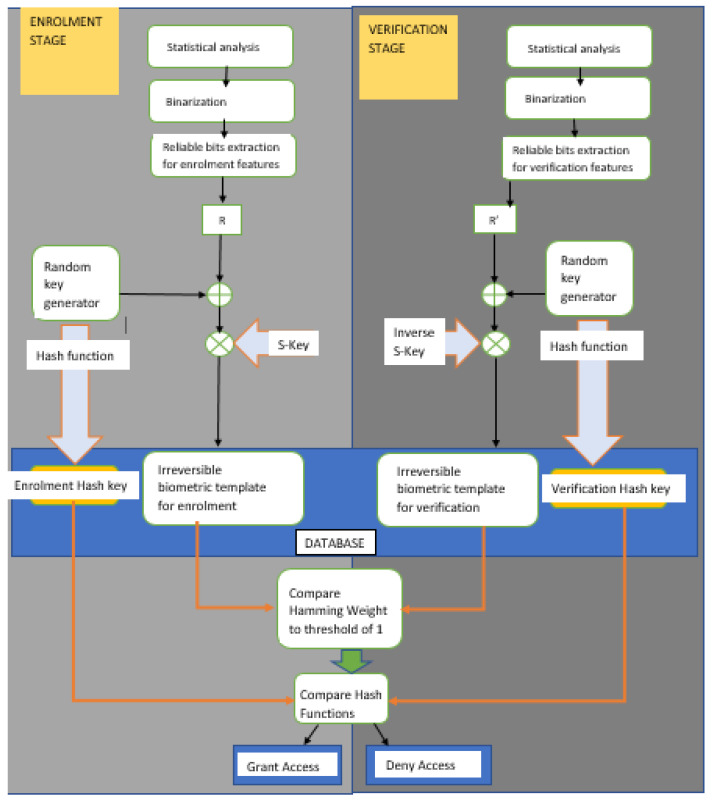
Fuzzy commitment flowchart.

**Table 1 entropy-24-00148-t001:** FRR and FAR for biometric cryptosystem for key sizes.

Message Length	Simulation Result	Mean	Min	Max
K = 5	FRR	25%	20%	50%
	FAR	20%	25%	40%
K = 7	FRR	15%	10%	30%
	FAR	10%	5%	25%
K = 11	FRR	30%	20%	50%
	FAR	35%	30%	55%

**Table 2 entropy-24-00148-t002:** Tenfold cross validation.

Classifier	Accuracy	Precision	True Positive	False Positive	ROC
K-Nearest Neighbor (IBK)	90%	0.72	0.71	0.02	0.89
Support vector Machine(SMO)	59%	0.62	0.59	0.02	0.95
KStar	46%	0.48	0.46	0.03	0.89
NaiveBayes	87%	0.93	0.93	0.02	0.86

## Data Availability

The data used in this work are available from the corresponding author upon reasonable request.

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
