# Peer review of "Object Selection as a Biometric"

_entropy, 2022, doi:10.3390/e24020148_

Round 1

Reviewer 1 Report

The manuscript deals with a very interesting topic of new aspects of the biometric technology. It offers a novel approach to this field. The Introduction section involves also the literature review. There is to note that the references mentioned in the manuscript are quite old regarding the novelty of the whole discussed issue. It is needed to comprise also the newer studies in order to get an actual picture of the proposed technology. The Materials and Methods section contain information about the data applied in the analysis. However, it is little bit misleading. What is meant as the pilot study? Is it the previous study done in the past? If yes, the reference is missing here. From a formal point of view, the subchapters titles are written in another font and indented variously. It is needed to unify it. The Results section reveal the quite impressive outcome. The receiver operating characteristic values are considerably high. Why accuracy of the support vector machine and KStar lower? K-Star is mentioned in Table 1 as “KStar”. Also, the other classifiers should be described in the methodology section. All the subchapters from 3.1 to the end of the Results section introduce only the methods and no results are involved here. Hence, it is needed to shift it into the Materials and Methods section. The Discussion section contains no reference and no comparison with the other studies.

Reviewer 2 Report

This paper describes a study on using eye movement as a biometric technique for authentication. The paper proposed various features such as fixation, dwell time, pupil diameter, etc to uniquely identify individuals. The idea is interesting, however, the paper doesn't provide sufficient details on how a real-world system would implement eye movement-based identification. Usually, authentication involved quick user feedback and it is unclear how eye movement will be used exactly to prove user identity. The experiment design and procedure are well described however the classification methods lack sufficient details. For example, What was the study on which the object selection was based? How was the data divided between test data and training data? How was the annotation done? What was the slack variable used in SVM? The classification methods used for the model training and testing need more details about the coefficients and control variables. Sections 3.1.6 to 3.2.2 are repeated twice and show a lack of proofreading prior to submission. The Figures also need better quality and improvement in terms of font size, consistency, color, flow clarity. The paper doesn't explain what BCH is and needs to use the expanded version of the term where it is first used in the text and needs to add a reference. Overall the idea seems promising however the manuscript does not do justice to it and needs to describe the experiment, analysis, and results in more depth to provide value to its readers. Therefore, the reviewer would recommend a major revision to improve the paper and resubmission.

Reviewer 3 Report

In this manuscript, the authors consider a technique for biometric authentication based on analyzing eye movement while selecting objects. They design an experiment in which they collect eye movement data of 20 users, then calculate some metrics used for authentication, build a model and establish a unique binary signature for each user. User authentication is done by classification an unknown sample to one of the prerecorded signatures.

My biggest concern with this manuscript is the poor and chaotic description of the method and the lack of explanation of the results obtained. 

The description of the results given in Table 1 is unclear. Which samples have you used to create the templates for the kNN method or training set for the remaining classifiers? And which ones have you selected for testing? How was the dataset containing 600 samples (30 for each user) divided while performing cross-validation? Did the testing set consist of three realizations for each user?

The manuscript structure is also chaotic. The results appear in the text before giving all necessary details about the classification process. 

Moreover, the description of the user enrollment and verification process is unclear. Figure 4 does not help understand it. Some blocks are too enigmatic, for example, statistical analysis, R.

In the sentence: 'A 93% accuracy rate was obtained although the performance of the model was poor.' what do you mean by the performance of the model? Is 93% accuracy a good result in the context of user authentication?

The description and the careless English do not convince me. 
In my opinion, the manuscript should be carefully rewritten. 

Round 2

Reviewer 1 Report

Many points are modified. However, the discussion section remains with no reference. It is not discussion itself. It looks like a summarisation of the findings only. It is needed to add references.

Author Response

Thank you very much for the feedback. A comparison and discussion of the findings was done. We compared our findings to the studies done by references [15],[20],[38] and [39].

Reviewer 2 Report

Thanks for the revised manuscript. The reviewers feedback and questions have been satisfactorily answered. 

Author Response

Thank you very much for the feedback.

Reviewer 3 Report

Thank you for the revised version and for addressing my comments and questions. I believe the manuscript has been significantly improved.

Author Response

Thank you very much for the feedback.